# Evolving Connectivity for Recurrent Spiking Neural Networks

**Guan Wang**[1, 2]*, **Yuhao Sun**[2, 3]*, **Sijie Cheng**[1, 4], **Sen Song**[2, 3]†
[1]Deptartment of Computer Science and Technology, Tsinghua University
[2]Laboratory of Brain and Intelligence, Tsinghua University
[3]Department of Biomedical Engineering, Tsinghua University
[4]Institute for AI Industry Research (AIR), Tsinghua University
`imonenext@gmail.com`, `{syh18,csj23}@mails.tsinghua.edu.cn`,
`songsen@tsinghua.edu.cn`

## Abstract

Recurrent spiking neural networks (RSNNs) hold great potential for advancing artificial general intelligence, as they draw inspiration from the biological nervous system and show promise in modeling complex dynamics. However, the widely-used surrogate gradient-based training methods for RSNNs are inherently inaccurate and unfriendly to neuromorphic hardware. To address these limitations, we propose the evolving connectivity (EC) framework, an inference-only method for training RSNNs. The EC framework reformulates weight-tuning as a search into parameterized connection probability distributions, and employs Natural Evolution Strategies (NES) for optimizing these distributions. Our EC framework circumvents the need for gradients and features hardware-friendly characteristics, including sparse boolean connections and high scalability. We evaluate EC on a series of standard robotic locomotion tasks, where it achieves comparable performance with deep neural networks and outperforms gradient-trained RSNNs, even solving the complex 17-DoF humanoid task. Additionally, the EC framework demonstrates a two to three fold speedup in efficiency compared to directly evolving parameters. By providing a performant and hardware-friendly alternative, the EC framework lays the groundwork for further energy-efficient applications of RSNNs and advances the development of neuromorphic devices. Our code is publicly available at `https://github.com/imoneoi/EvolvingConnectivity` .

## 1 Introduction

Inspired by the remarkable information processing power of the human brain, neuromorphic computing aims to design and implement computational systems that mimic the architecture and functionality of biological neural networks, paving the way for a new era of intelligent machines. Specifically, the highly recurrently connected and spike-emitting brain network inspired recurrent spiking neural networks (RSNNs), which employ discrete, spike-based signals for transmitting information through recurrent connections in an event-driven manner. RSNNs can serve as realistic models of the brain incorporating the latest anatomical and neuro-physiological data [Billeh et al., 2020], leading to the discovery of general principles of computing, e.g., noise robustness [Chen et al., 2022]. Furthermore, RSNNs can exhibit self-organized critical properties [Poil et al., 2012] and serve as a reservoir for complex temporal dynamics [Jaeger, 2001, Maass et al., 2002], making them suitable for sequential tasks like robotics [Lele et al., 2020], path-finding [Rueckert et al., 2016], and others.

---

*Equal contribution
†Corresponding author

37th Conference on Neural Information Processing Systems (NeurIPS 2023).

Despite the importance of RSNNs, developing end-to-end training algorithms remains a challenge. Inspired by the success of deep learning, a line of research [Wu et al., 2018, Shrestha and Orchard, 2018, Bauer et al., 2022] uses error-backpropagation [Rumelhart et al., 1985] with carefully chosen surrogate gradients to address the non-differentiability problem and achieve performance comparable to deep learning. However, incorporating surrogate gradients into RSNNs introduces two concerns. Algorithmically, the surrogate gradient leads to inherent inaccuracy in the descent direction [Li et al., 2021] and sensitivity to function scale selection [Zenke and Vogels, 2021]. At the implementation level, gradient-based training is incompatible with prominent neuromorphic devices [Pei et al., 2019, Davies et al., 2018, Mayr et al., 2019] due to the requirement of accessing the full network state over every timestep [Werbos, 1990]. Consequently, this raises a critical research question: *can we design a training method for RSNNs that bypasses the need for gradients without compromising performance?*

In response to this challenge, we observe that connection probability distribution information from brain connection maps across various species exhibit similarities [Haber et al., 2023] and can be used to construct large-scale neural computing models [Billeh et al., 2020, Schmidt et al., 2018]. Existing studies demonstrate that networks with binary connections alone can achieve high performance, rivaling that of weighted networks [Gaier and Ha, 2019, Frankle and Carbin, 2018, Malach et al., 2020]. Moreover, computing boolean connections relies primarily on integer arithmetic rather than resource-intensive floating-point operations, enabling simpler on-chip implementation and enhanced energy efficiency. Considering these features, we reformulate the architecture of RSNNs, where connections are sampled independently from parametric Bernoulli distributions, and adopt Natural Evolution Strategies (NES) [Wierstra et al., 2014] to optimize this parametric probability distribution, providing a scalable, inference-only, and effective algorithm for tuning parameters.

In this paper, we introduce the evolving connectivity (EC) framework for training RSNNs. The EC framework reformulates RSNNs as boolean connections with homogeneous weights and employs NES to evolve the connection probabilities of the network. To evaluate effectiveness and efficiency, we conduct extensive experiments on locomotion tasks, a set of widely-used sequential decision-making tasks [Brockman et al., 2016, Freeman et al., 2021]. Experimental results show that our proposed EC method achieves performance comparable to deep recurrent neural networks, surpassing the asymptotic performance of surrogate gradients. Despite using a GPU, which is not specifically optimized for our framework, the EC method yields a speed improvement of $2 \sim 3\times$ compared to directly evolving parameters using Evolution Strategies [Salimans et al., 2017] and exhibits better efficiency than surrogate gradients. Additionally, our EC method can effectively train deep recurrent neural networks (RNNs), demonstrating versatility for different architectures and potential for quantized deep neural networks.

Our main contributions are summarized as follows:

- **Novel framework**: We propose a novel inference-only training framework for RSNNs by reformulating weight-tuning as connection probability searching through evolution-based algorithms.
- **High performance**: Our method can solve the complex 17-DoF humanoid locomotion task with RSNN, achieving performance on par with recurrent neural networks and outperforming gradient-trained RSNNs.
- **Hardware-friendly**: By producing RSNNs with sparse 1-bit boolean connections, as well as enabling inference-only training and high scalability, our method is highly compatible with neuromorphic devices. This compatibility holds promising potential for further energy-efficient applications of RSNNs.

## 2 Related Works

### 2.1 Training Recurrent Spiking Neural Networks

The study of training algorithms for RSNNs endeavors to unravel the mechanism that enables the brain to learn. For decades, biological plasticity rules, especially spike-timing-dependent plasticity (STDP) [Bi and ming Poo, 1998], have been considered as a foundational of RSNNs training [Diehl and Cook, 2015, Kheradpisheh et al., 2018, Mozafari et al., 2019]. Recently, with the success of deep learning, surrogate gradient-based approaches [Wu et al., 2018, Shrestha and Orchard, 2018, Bauer et al., 2022] have been predominately utilized in training RSNNs. Surrogate gradients share conceptual similarities with the straight-through estimator [Bengio et al., 2013] used in deep learning.

Both techniques employ continuous surrogate functions to compute gradients of non-differentiable functions during the backward pass, while preserving the original non-differentiable functions in the forward pass. Slayer [Shrestha and Orchard, 2018] adopted an exponential decaying surrogate function with a spike response model (SRM), while STBP [Wu et al., 2018] incorporated BPTT in multi-layer SNN training and proposed several surrogate function choices. Chen et al. [2022] trained an anatomical and neurophysiological data constrained RSNN and demonstrated its robustness and versatility. However, gradient-based approaches face the new problem of being difficult to implement on neuromorphic devices. To address the limitation, E-prop [Bellec et al., 2020] utilize eligibility trace to compute truncated gradient, becoming an exemplary training algorithm on neuromorphic devices like Loihi2 [Davies et al., 2018] and SpiNNaker2 [Mayr et al., 2019], but have a strict restriction on the temporal relationship of SNN models and still lags behind surrogate gradient methods in performance. Our approach tackles the gradient-effectiveness dilemma from the beginning, by introducing the inference-only training framework.

## 2.2 Weight-agnostic Neural Networks

Training a neural network typically involves assigning appropriate values to the network's weights. However, the weight-agnostic neural network (WANN) [Gaier and Ha, 2019] has demonstrated that a network's topology can be highly informative, similar to that of a weighted neural network. Additionally, research on the lottery ticket hypothesis (LTH) [Frankle and Carbin, 2018, Zhou et al., 2019, Ramanujan et al., 2020] suggests that an over-parameterized network contains an effective subnetwork, even when using its initial parameter weights. Several theoretical studies have also shown that a sufficiently large, randomly-weighted deep neural network contains a subnetwork capable of approximating any function [Malach et al., 2020, Fischer et al., 2022]. Our approach, furthering the existing research on connection-encoded networks, proposes a framework that utilizes connection probability to parameterize the network.

## 2.3 Deep Neuroevolution

Deep neuroevolution utilizes evolutionary algorithms for training deep neural networks. Natural Evolution Strategies (NES) [Wierstra et al., 2014] have laid the foundation for gradient estimation in this domain. A well-known variant, Evolution Strategies (ES) [Salimans et al., 2017], has been employed for training deep neural networks in sequential decision-making tasks. ES optimizes a single set of continuous network weights by applying Gaussian perturbations and updating the weights using the NES gradient estimator. Following the success of ES, numerous evolutionary algorithms have been proposed for training deep neural networks. For example, Such et al. [2018] showed that deep neural networks could be effectively trained using simple genetic algorithms that mutate weights. Furthermore, Conti et al. [2018] integrated exploration and novelty-seeking into ES to overcome local minima. While the majority of prior research has primarily focused on continuous parameters, our proposed framework presents a novel approach by concentrating on the search for connection probability distributions. This shift in perspective offers new possibilities for evolving recurrent spiking neural networks in a hardware-friendly manner.

## 3 Preliminaries: Recurrent Spiking Neural Networks

RSNN is a class of spiking neural networks which incorporate feedback connections. In this paper, we adopt a typical RSNN architecture from the reservoir network [Jaeger, 2001, Maass et al., 2002] for sequential tasks. It is worth noting that, despite that we adopt a specific RSNN model as an example, our framework can be broadly applied to search for connectivity distributions in any type of RSNN, as it does not depend on network-specific assumptions and only relies on evaluating the network with a set of parameters.

According to Dale's law, our network consists of an excitatory neuron group and an inhibitory neuron group. Each neuron is modeled as a leaky integrate and fire (LIF) neuron. A neuron will fire a spike when its membrane potential $u$ exceeds the threshold, and the membrane potential will hard-reset to $0$. More specifically, our model defines the dynamics of membrane potential $u$ and synaptic current $c$ as follows:

$$\tau_m \frac{\mathrm{d}\mathbf{u}^{(g)}}{\mathrm{d}t} = -\mathbf{u}^{(g)} + R\mathbf{c}^{(g)} \tag{1}$$

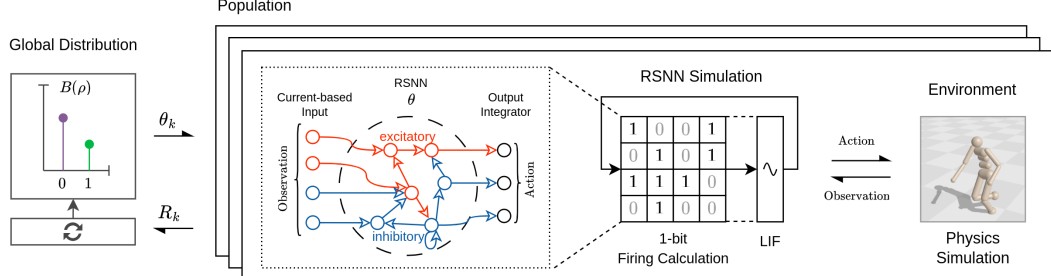

Figure 1: Architecture of evolving connectivity (EC). The connectivity $\boldsymbol{\theta}_k$ of the population is sampled from the global distribution $B(\boldsymbol{\rho})$ and then evaluated in parallel. The RSNN consists of excitatory and inhibitory neurons, simulated using 1-bit firing calculations and the LIF model.

where $g = \{Exc, Inh\}$ denotes the excitatory and inhibitory group, respectively, and $R$ denotes the resistance.

The current input was modeled using exponential synapses to retain information for a short duration, commonly adopted in robotic tasks [Tang et al., 2020, Naya et al., 2021].

$$\frac{d\mathbf{c}^{(g)}}{dt} = -\frac{\mathbf{c}^{(g)}}{\tau_{syn}} + \sum_{g_j} I_{g_j} \sum_j \mathbf{W}_{ij}^{(g_i g_j)} \delta(t - t_j^{s(g_j)}) + \mathbf{I}_{ext} \tag{2}$$

where $t_j^{s(g_j)}$ denotes the spike time of neuron $j$ in neuron group $g_j$, $\delta$ denotes Dirac delta function. $I_g$ was set to define the connection strength of excitatory and inhibitory synapse respectively, where $I_{Exc} > 0$ and $I_{Inh} < 0$. Weight $\mathbf{W}^{(g_i g_j)}$ was defined as a matrix with non-negative elements, connecting group $g_j$ to group $g_i$. Besides, the external input signal $\mathbf{I}_{ext}$ is extracted using linear projection of observation $\mathbf{x}$.

Discretizing the LIF differential equations (Eq. 1 and 2) with $\Delta t$ as a time-step, we could obtain the following difference equation for our RSNN model:

$$\mathbf{c}^{(t,g)} = d_c \mathbf{c}^{(t-1,g)} + \sum_{g_j} I_{g_j} \mathbf{W}^{(g_i g_j)} \mathbf{s}^{(t-1,g_j)} + \mathbf{I}_{ext}^{(t,g)} \tag{3}$$

$$\mathbf{v}^{(t,g)} = d_v \mathbf{u}^{(t-1,g)} + R\mathbf{c}^{(t,g)} \tag{4}$$

$$\mathbf{s}^{(t,g)} = \mathbf{v}^{(t,g)} > \mathbf{1} \tag{5}$$

$$\mathbf{u}^{(t,g)} = \mathbf{v}^{(t,g)}(\mathbf{1} - \mathbf{s}^{(t,g)}) \tag{6}$$

where $d_c = e^{-\frac{\Delta t}{\tau_{syn}}}$ and $d_v = e^{-\frac{\Delta t}{\tau_m}}$ are two constant parameters.

The output vector $\mathbf{o}$ is extracted using linear projection of neuron firing rates in a short time period $\tau$, i.e.,

$$\mathbf{o}^{(t)} = \sum_\tau k(\tau) \sum_g \mathbf{W}_{out}^{(g)} \mathbf{s}^{(t-\tau,g)} \tag{7}$$

where $\mathbf{W}_{out}^{(g)}$ denotes the output weight, and $k$ is a function averaging the time period $\tau$.

## 4 Framework

In this section, we present our proposed Evolving Connectivity (EC) framework, which is depicted in Fig. 1. Our approach consists of three main steps: (1) reformulating the neural network architecture from weight-based parameterization to connection probability distribution, (2) employing the Natural Evolution Strategies (NES) method to optimize the reformulated parameter space, and (3) deterministically extracting the final parameters from the distribution.

**Reformulation.** A sparse connected weight matrix can be described into a weight matrix $\mathbf{w}$ and a connection mask $\boldsymbol{\theta}$. Traditionally, we can use Erdős–Rényi random matrix to describe the connection

mask, in which connections are independently drawn from a Bernoulli distribution, i.e.,

$$\mathbf{W}_{ij} = \mathbf{w}_{ij} \cdot \boldsymbol{\theta}_{ij}, \text{where } \boldsymbol{\theta}_{ij} \sim B(\rho) \tag{8}$$

Inspired by the success of subnetworks in deep neural networks [Ramanujan et al., 2020], we can reformulate this in a connection framework. In this view, we aim to find a connection probability matrix, $\boldsymbol{\rho} = (\rho_{ij})$, where each element represents the connection probability between two neurons, and set all weights $\mathbf{w}_{ij}$ to be unit size, i.e.,

$$\mathbf{W}_{ij} = \boldsymbol{\theta}_{ij}, \text{where } \boldsymbol{\theta}_{ij} \sim B(\boldsymbol{\rho}_{ij}). \tag{9}$$

**Optimization.** We optimize $\boldsymbol{\rho}$ to maximize the expected performance metric function $R(\cdot)$ across individual network samples drawn from the distribution, which can be expressed as the following objective function:

$$\boldsymbol{\rho}^* = \arg\max_{\boldsymbol{\rho}} J(\boldsymbol{\rho}) = \arg\max_{\boldsymbol{\rho}} \mathbb{E}_{\boldsymbol{\theta} \sim B(\boldsymbol{\rho})}[R(\boldsymbol{\theta})] \tag{10}$$

To optimize this objective, we employ Natural Evolution Strategies (NES) [Wierstra et al., 2014]. NES provides an unbiased Monte Carlo gradient estimation of the objective $J(\boldsymbol{\rho})$, by evaluating the performance metric $R_k$ on multiple individual samples $\boldsymbol{\theta}_k$ drawn from $B(\boldsymbol{\rho})$. Specifically, NES estimates the gradient of $J(\boldsymbol{\rho})$ with respect to the parameters of the distribution $\boldsymbol{\rho}$ by computing the expectation over samples from $B(\boldsymbol{\rho})$:

$$\nabla_{\boldsymbol{\rho}} J(\boldsymbol{\rho}) = \mathbb{E}_{\boldsymbol{\theta} \sim B(\boldsymbol{\rho})}[\nabla_{\boldsymbol{\rho}} \log P(\boldsymbol{\theta}|\boldsymbol{\rho})R(\boldsymbol{\theta})] \tag{11}$$

$$= \mathbb{E}_{\boldsymbol{\theta} \sim B(\boldsymbol{\rho})}\left[\frac{\boldsymbol{\theta} - \boldsymbol{\rho}}{\boldsymbol{\rho}(1-\boldsymbol{\rho})}R(\boldsymbol{\theta})\right] \tag{12}$$

$$\approx \frac{1}{N}\sum_{k=1}^{N}\frac{\boldsymbol{\theta}_k - \boldsymbol{\rho}}{\boldsymbol{\rho}(1-\boldsymbol{\rho})}R_k \tag{13}$$

Thus, we obtained an inference-only estimation of the gradient by simply sampling and evaluating the metric. Then gradient descent can be carried over the estimated gradients, following the NES approach [Wierstra et al., 2014]. Furthermore, as in Williams [1992], we scale the step size proportionally to the variance, i.e. $\alpha = \eta \cdot \text{Var}[B(\boldsymbol{\rho})]$, and obtain the update rule:

$$\boldsymbol{\rho}_t = \boldsymbol{\rho}_{t-1} + \alpha \nabla_{\boldsymbol{\rho}} J(\boldsymbol{\rho}) \approx \boldsymbol{\rho}_{t-1} + \frac{\eta}{N}\sum_{k=1}^{N}(\boldsymbol{\theta}_k - \boldsymbol{\rho})R_k \tag{14}$$

In addition, all elements of $\boldsymbol{\rho}$ clipped in the interval $[\epsilon, 1-\epsilon]$, where $\epsilon \to 0^+$, to guarantee a minimal level of exploration within the search space. In our experiment, we set $\epsilon = 0.001$ and initialize $\boldsymbol{\rho} = 0.5$. Additionally, we adopted a similar fitness shaping trick, center rank transform, as discussed in Salimans et al. [2017], which sort the return of a population, then linearly rescale the sorting indexes into a fixed interval $[-0.5, 0.5]$.

Once a sufficient number of updates have been performed, the final parameter $\boldsymbol{\theta}$ can be deterministically obtained as the parameter with maximum probability, which is subsequently used for deployment. In the context of a Bernoulli distribution, this process is equivalent to thresholding $\boldsymbol{\rho}$ at 0.5 to produce a boolean matrix containing only $\{0, 1\}$ values.

## 5 Properties of EC Framework

In this section, we further discuss the important properties of our proposed EC framework in implementation, thanks to leveraging the connection probability distribution.

**Inference only.** The most significant challenge in neuromorphic devices is the absence of an effective hardware-friendly learning algorithm [Li et al., 2023]. Most neuromorphic chips, such as Loihi [Davies et al., 2018], Tianjic [Pei et al., 2019], TrueNorth [Akopyan et al., 2015], SpiNNaker2 [Mayr et al., 2019], lack support for directly calculating error backpropagation, which is crucial for surrogate gradient methods. The inference-only EC framework enables an alternative method for training on these inference chips.

**1-bit connections.** EC employs 1-bit sparse connections throughout training and deployment, replacing the traditional floating-point weight matrix. Therefore, the 1-bit connections permit the use of more economical integer arithmetic instead of costly floating-point cores. This approach not only accelerates computations on devices like GPUs, but also holds a promise of driving the creation of novel 1-bit connection neuromorphic computing hardware.

**Scalability.** The inference-only property of the EC framework implies no data dependence between evaluations, which allows for the distribution of sampled parameters $\theta_k$ across independent workers and the collection of their reported performance measures, making EC highly scalable. Moreover, the random seed for sampling the population can be transmitted across nodes instead of the connections $\theta_k$ to minimize communication overhead, facilitating scalar-only communication.

## 6 Experiments

### 6.1 Experimental Setups

**Tasks.** We focus on three robotic locomotion tasks in our experiments, Humanoid, Walker2d, and Hopper, as they are commonly used for sequential decision-making problems in the reinforcement learning domain [Brockman et al., 2016, Freeman et al., 2021]. As illustrated in Fig.

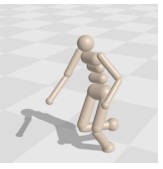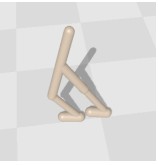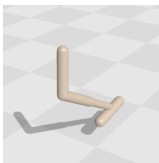

Figure 2: Locomotion tasks illustrated from left to right: Humanoid (17-DoF), Walker2d (6-DoF), and Hopper (3-DoF).

2, these tasks involve controlling robots with varying degrees of freedom (DoF), to perform certain actions that maximize the return within a fixed-length episode $T$. In the EC framework, the performance metric function can be defined as the expected return over episodes $R(\theta) = \mathbb{E}_{\tau \sim \pi_\theta}[\sum_{t=0}^{T} r_t]$, and a single evaluation $R_i$ corresponds to the return of one episode using network parameter $\theta_k$.

**RSNN architecture.** In our study, we employed a RSNN consisting of an input layer, a hidden recurrent layer with LIF neurons, and an output layer, as delineated in the Preliminaries section. To achieve an equilibrium within the excitatory inhibitory balance, input and hidden layers were evenly divided into excitatory and inhibitory group.

**Baselines.** To thoroughly evaluate the effectiveness of our framework and its advantages over prior methods, we compare our EC framework with deep RNNs, as well as RSNN trained with Surrogate Gradients (SG) and Evolution Strategies (ES). Detailed network architecture and precision is illustrated in Table 1.

For deep RNNs, we employ widely-used recurrent deep neural networks, specifically long-short term memory (LSTM) [Hochreiter and Schmidhuber, 1997] and gated recurrent unit (GRU) [Cho et al., 2014], trained using Evolution Strategies (ES) [Salimans et al., 2017] as our baselines.

For RSNN, we employ the same structured but excitatory/inhibitory not separated RSNN trained with ES and Surrogate Gradient (SG). ES is directly applied to search the weight matrix $\mathbf{W}$ of RSNNs, optimizing the performance metric in a gradient-free manner. In contrast, the Surrogate Gradient is combined with Proximal Policy Optimization (PPO) [Schulman et al., 2017], a prominent reinforcement learning method, to optimize the weight $\mathbf{W}$ for sequential decision-making tasks where returns cannot be differentiated.

Table 1: Comparison of model architectures.

| Model | Hidden size | # Params | Precision | Size (MB) |
|-------|-------------|----------|-----------|-----------|
| RSNN | 256 | 193K | 1-bit(EC) / FP32(Others) | 24KB(EC) / 768KB(Others) |
| GRU | 256 | 386K | FP32 | 1544KB |
| LSTM | 128 | 191K | FP32 | 764KB |

**Implementation Details.** Our EC framework and all baselines are implemented using the JAX library [Bradbury et al., 2018] and just-in-time compiled with the Brax physics simulator [Freeman et al., 2021] for efficient GPU execution. The population is vectorized automatically, and 1-bit firing

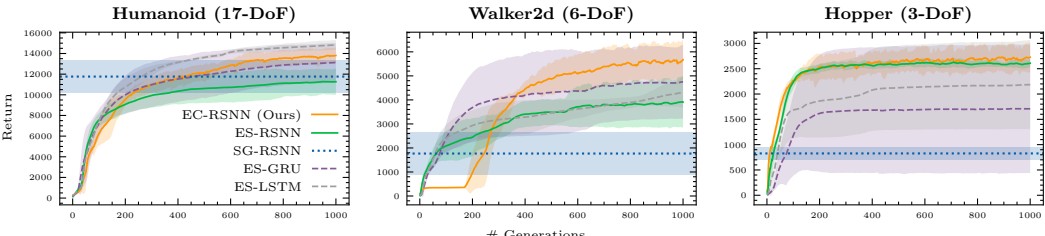

Figure 3: Performance evaluation on locomotion tasks. SG-RSNN is plotted based on the final return, as gradient-based updates are not directly comparable to generations in evolution-based methods. The proposed Evolving Connectivity (EC) framework effectively solves the 17-DoF Humanoid locomotion task, demonstrating competitive performance with deep RNNs and outperforming RSNNs trained using both Surrogate Gradient and Evolution Strategies across all tasks.

calculations take advantage of INT8 arithmetic for performance optimization. As a result, the training process achieves over $180,000$ frames per second on a single NVIDIA TITAN RTX GPU. Each experiment's result is averaged over 3 independent seeds, with the standard deviation displayed as a shaded area. For detailed information on hyperparameters and hardware specifications, please refer to Appendix G.

## 6.2 Performance Evaluation

Firstly, we compare RSNN trained with our EC framework (EC-RSNN) to deep RNNs. To ensure a fair comparison, we utilized the evolution-based training approach ES [Salimans et al., 2017] for deep RNNs, including ES-GRU and ES-LSTM. Typically, SNNs face challenges in surpassing the performance of their DNN counterparts. However, as shown in the experimental results in Fig. 3, our EC-RSNN achieves competitive performance and training efficiency compared to deep RNNs, even surpassing ES-GRU and ES-LSTM on the Walker2d and Hopper tasks. In the complex 17-DoF Humanoid task, our EC-RSNN could also outperform ES-GRU. Moreover, EC-RSNN demonstrates comparable performance to GRU and LSTM trained with PPO, as detailed in Appendix E. It is worth noting that, while GRU and LSTM use densely connected, full precision (32-bit floating-point) weights and hard-coded gating mechanisms, EC-RSNN is constrained on 1-bit sparse weights and adheres to Dale's law for weight signs.

Then, as for the same structured RSNNs, our EC-RSNN outperforms ES-RSNN, especially significantly surpassing ES-RSNN on complex Humanoid and Walker2d tasks. We highlight two aspects that potentially account for the superior performance of our EC framework. On the one hand, schema theory [Whitley, 1994] suggests that selecting over a population of binary strings could combine partial binary string patterns (schemata) to implicitly work on many solutions without evaluating them explicitly, leading to massive parallelism. EC leverages the discrete $\{0, 1\}$-string space, which may enable more efficient optimization due to implicit parallelism. On the other hand, ES uses fixed-scale noise to perturb a set of deterministic parameters, which may often fall in wide areas in the landscape of the objective function and fail to proceed on sharp maxima [Lehman et al., 2018]. In contrast, EC operates over a probabilistic distribution, providing the flexibility to adjust variance on sensitive parameters and achieving more fine-grained optimization.

Finally, gradient updates cannot be directly converted into evolution generations, thus we only compare the final converged return for surrogate gradient trained RSNNs. Considering that the surrogate gradient approach is sensitive to the selection of the surrogate function and its parameters [Zenke and Vogels, 2021], we choose three commonly-used surrogate functions with different $\beta$ and $\gamma$ parameters to thoroughly validate the performance: SuperSpike [Zenke and Ganguli, 2018], the piecewise linear function proposed by Esser et al. [2016], and the derivative of the Sigmoid function [Zenke and Ganguli, 2018]. As shown in Fig. 4, the performance of our EC-RSNNs is consistently higher than SG-RSNNs across all surrogate functions and hyper-parameters on the complex 17-DoF Humanoid task. We adopt the best parameter set of the piecewise linear surrogate function, with $\gamma = 0.8$, $\beta = 2$ as the baseline in Fig. 3.

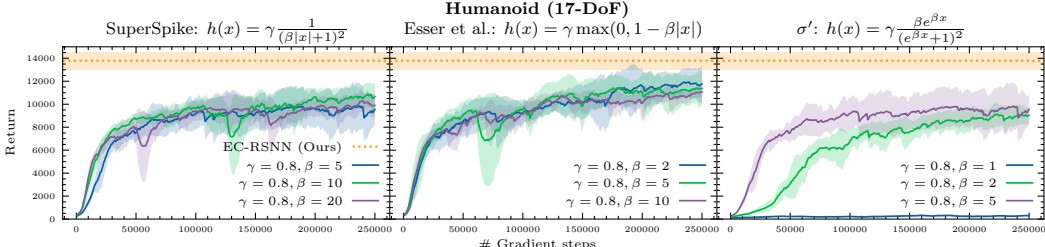

Figure 4: Surrogate Gradient on the Humanoid task, with $\gamma = 0.8$ and several surrogate functions, $\beta$ parameters. For $\gamma = 1.0$ results, please refer to Appendix F. Performance is sensitive to the chosen surrogate function and its parameters. In this context, EC surpass all considered parameters of SG.

One possible reason to explain the performance of SG-RSNN is that due to RSNNs having both explicit and implicit recurrence, the choice of surrogate function may determine whether gradients vanish or explode [Zenke and Vogels, 2021]. Moreover, the gradient approximation of SG is inaccurate and may deviate from the steepest descent direction [Li et al., 2021]. In contrast, our EC framework leverages unbiased Monte-Carlo gradient estimation, which approaches the optimal direction given that the population is sufficiently large [Zhang et al., 2017].

## 6.3 Efficiency Comparison

In this section, we compare the computational efficiency of our proposed EC framework with other baselines, as shown in Fig. 5. To ensure a fair comparison, we evaluate the EC and baseline methods using the same wall-clock computation time and implement them on identical hardware.

As expected, EC-RSNN exhibits slower training than deep RNNs due to the higher complexity of RSNNs, which necessitates simulating multiple timesteps to compute firing and integrate differential equations. Additionally, the deep RNN models employed in this study, specifically LSTM and GRU, are well-established and highly optimized for GPU implementation. Nevertheless, within the same computation time, the performance achieved by EC-RSNN remains competitive with deep RNNs.

Furthermore, we execute both ES and EC for 1,000 generations using the same population size and RSNN architecture. Experimental results in Fig. 5 have shown that our 1-bit EC-RSNN achieves a speedup of approximately $2 \sim 3\times$ over 32-bit floating-point ES-RSNN. This finding emphasizes the value of 1-bit connections within the proposed framework. Utilizing integer arithmetic typically results in higher throughput than floating-point arithmetic across most accelerators, and incorporating smaller data types reduces both memory requirements and memory access time. It is also worth mentioning that the 1-bit connections are implemented using INT8 on GPU, which is not fully optimized. By implementing the framework on hardware supporting smaller data types, connection size could be further reduced by up to $8\times$, enabling more significant acceleration and cost reduction.

Additionally, our EC-RSNN demonstrates faster convergence compared to SG-RSNN. This can be attributed to the fact that EC-RSNN is an inference-only method with 1-bit connections, requiring

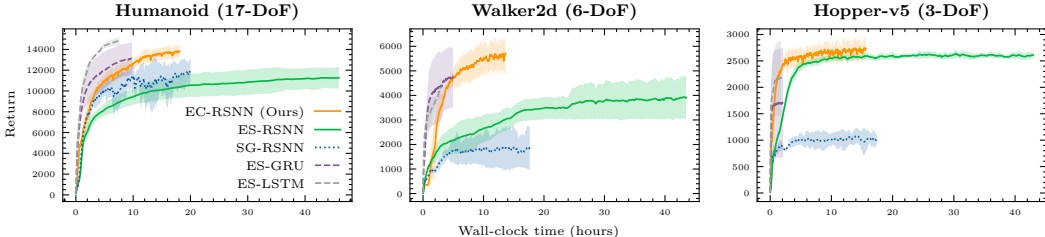

Figure 5: Efficiency comparison in locomotion tasks. Evolution-based approaches (EC, ES) are executed for 1,000 generations, while surrogate gradient (SG) runs for 250,000 gradient steps to attain a comparable run-time to EC, providing a fair comparison. When training RSNNs, EC attains a $2 \sim 3\times$ speedup over ES and demonstrates faster convergence than SG.

only a single forward pass using integer arithmetic, while SG-RSNN demands both a forward and a backward pass employing computationally-intensive floating-point operations.

Finally, we calculated the estimated energy consumption of the EC-RSNN on both GPU and neuromorphic devices, aiming to assess the power efficiency of the EC to train RSNN. In our study, all experiments were consistently conducted on the same NVIDIA Titan RTX GPU, operating at a stable 100% GPU power (280W). Consequently, the energy consumption of the EC training process on GPU is approximated to be directly proportional to the computation wall time, resulting in a 9 MJ. For neuromorphic device, we have conducted computations pertaining to the estimated energy consumption of EC-RSNN when implemented on the Loihi chip [Davies et al., 2018]. As detailed in Appendix D, the estimated energy consumption of EC is 28 kJ, which indicate a noticeable reduction in power consumption by more than two orders of magnitude.

### 6.4 General 1-bit Framework

The task of discretizing a neural network into a 1-bit representation presents a considerable challenge, particularly when employing continuous training techniques such as ES and SGD. To illustrate this difficulty, we attempted to discretize both ES and SG for training 1-bit connection RSNNs utilizing the straight-through estimator [Bengio et al., 2013]. Fig. 7 (a) showed that while both ES-RSNN (1bit) and SG-RSNN (1bit) exhibited learning progress, they were surpassed by EC-RSNN (1bit). Fig. 7 (b) and (c) suggest that ES and SG demonstrated superior performance on continuous FP32 weights as opposed to discrete 1-bit representations. These findings imply that continuous optimization methods excel when applied to continuous parameters; conversely, for 1-bit discrete connections, EC is recommended

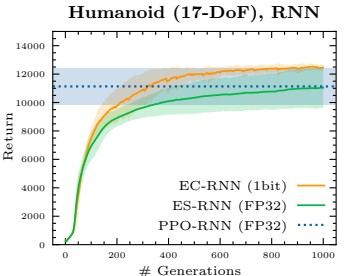

Figure 6: Performance comparison of vanilla RNN trained with EC, ES, PPO. EC surpasses baselines despite using 1-bit connections.

due to its specific design for discrete 1-bit optimization and the provision of unbiased gradients. A comprehensive description of the discretization for ES and SG methods is provided in Appendix C.

Moreover, Despite EC have demonstrated the efficacy on RSNNs, it serves as a versatile 1-bit training framework to non-spiking networks as well. To substantiate EC's potential in training deep RNN, we conducted a series of experiments employing a standard RNN model trained with EC, while using ES and PPO as baselines. The RNN has 256 tanh units in the hidden layer. For 1-bit EC, the weight magnitudes are 0-1 connection matrix, and the weight signs are separated to excitatory or inhibitory. The results in Fig. 6 demonstrate that EC can effectively train 1-bit deep recurrent neural networks and has the potential for different architectures and quantized neural networks.

## 7 Conclusion

In this study, we present the innovative Evolving Connectivity (EC) framework, a unique inference-only approach for training RSNN with 1-bit connections. The key attributes of the EC framework,

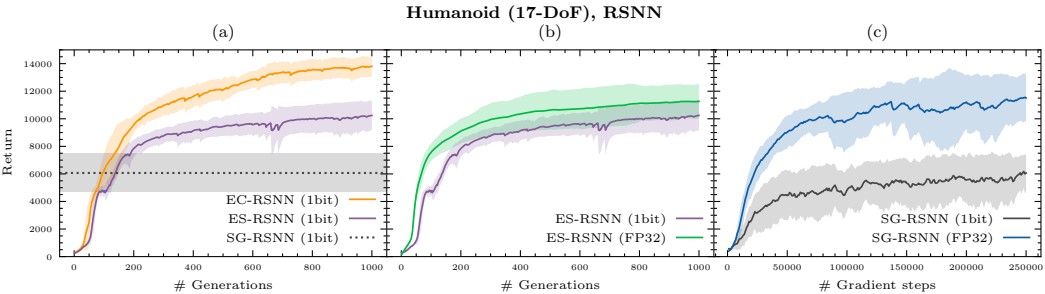

Figure 7: Optimization of the 1-bit RSNN proposed in this study using ES and SG. (a) In the discrete 1-bit RSNN settings, EC significantly outperforms ES and SG. (b) (c) Comparison of FP32 and 1-bit precision for ES and SG respectively, indicating that these continuous optimization methods achieve optimal results with FP32 continuous weights.

such as its inference-only nature and scalability, render it particularly suitable for training RSNNs on neuromorphic devices. The use of 1-bit connections significantly reduces memory requirements and computational cost, facilitating faster training on GPUs and paving the way for additional cost reductions in neuromorphic chip production.

We performed extensive experiments on a variety of intricate locomotion tasks, showcasing the competitive performance of our proposed EC framework in comparison to deep RNNs like LSTM and GRU. Moreover, the EC framework outperforms other RSNN training methods such as Evolution Strategies and Surrogate Gradient, both in terms of performance and computational efficiency.

## 8  Limitations

Evolutionary algorithms demand the storage of $N$ distinct parameter sets for population evaluation, leading to a space complexity of $\mathrm{O}(N|\theta|)$. In contrast, gradient-based approaches utilize a single parameter set but necessitate the storage of intermediate results at each timestep, resulting in a space complexity of $\mathrm{O}(NHS + |\theta|)$, where $H$ corresponds to the number of BPTT timesteps, and $S$ represents the size of the intermediate results. This situation gives rise to a trade-off: evolutionary methods offer greater memory efficiency for tasks featuring long time horizons, whereas gradient-based techniques with larger parameter sizes and shorter time horizons require less memory. Recognizing this trade-off is crucial in practical applications. Although our proposed EC framework, as an evolutionary algorithm, retains the same space complexity and trade-off, it can achieve a constant term reduction by storing 1-bit connections.

## 9  Discussions

**Neuromorphic hardware.** One primary bottleneck in the development of neuromorphic hardware is the lack of effective on-chip learning algorithms to build applications compared to deep learning approaches. This paper proposes a novel EC framework that circumvents the requirement for gradients and demonstrates efficacy on locomotion tasks, providing a potential solution for solving the on-chip learning challenge. Typically, the demand for computing devices falls into two categories, including cloud and edge. In the cloud, our proposed EC framework supports large-scale learning, while at the edge, it enables energy-efficient applications. Therefore, our framework offers a potential approach to further building neuromorphic applications.

Another practical challenge lies in the trade-off between numeric precision and cost. We suggest that it is possible to drastically reduce connections from floating point to 1-bit, providing a novel design principle for the next generation of neuromorphic hardware. Therefore, our method holds the potential to significantly decrease the manufacturing and energy costs of neuromorphic hardware.

**Neuroscience.** Our EC framework introduces a novel method for neuroscience research. First, as opposed to toy tasks or simulated signals, which are often studied in previous neuromodeling work, we employed RSNN in a complex, real-world like locomotion task. Additionally, controlling signals in neuroscience like 'GO' and 'NOGO' signals in the basal ganglia can be easily integrated into the task by concatenating them to the environment observation vector, enabling the creation of novel neuroscientific tasks. Our work lays the foundation for further investigation of decision-making processes and motor control.

Moreover, our framework provides a novel type of data to analyze: neuron-to-neuron level connection probability. Connection probability is one of the most fundamental properties in brain-wide connectomes. However, obtaining neuron-to-neuron connection probability from the whole mammalian brain is experimentally implausible due to limitations in nowadays neuroconnectomic technology. Our work provides in silico connection data for further analysis, including covariances, motifs, clustered engrams, and dimensional properties.

Finally, our framework is capable of incorporating neuroanatomical and neurophysiological data to construct a novel neurosimulation model, since our framework is able to train arbitrary models in principle. Neuroscientists have discovered that connection probability between two neurons is determined by various factors, including spatial distance, receptive field, and neuron types. Our framework can leverage these findings and conduct in silico experiments with data-driven models.

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

## A    Acknowledgement

This work was supported by the National Key Research and Development Program of China (grant 2021ZD0200301).

## B    Comparing EC and NES

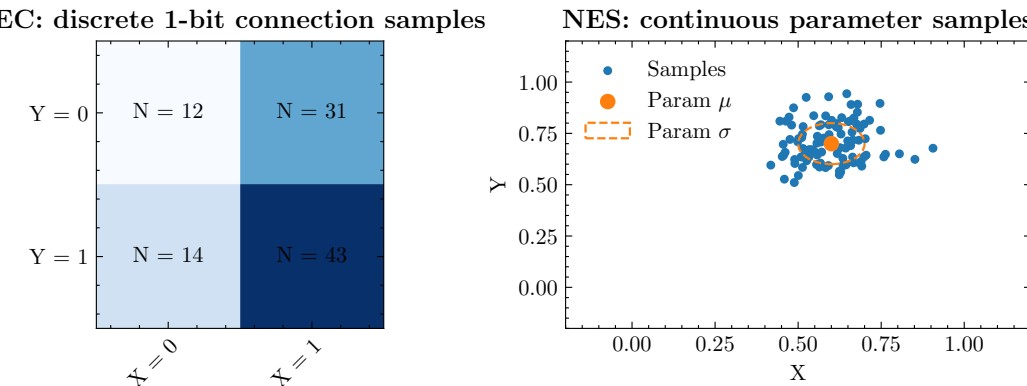

Figure S1: Comparison of search spaces in EC and NES with $N = 100$ samples and two parameters $(X, Y)$. EC samples from a discrete 1-bit search space, while NES samples from a continuous search space.

For readers familiar with evolutionary algorithms, we present the similarities and main differences between our Evolving Connectivity (EC) method and commonly used evolutionary algorithms, namely Natural Evolution Strategies (NES, Wierstra et al. [2014]) and Evolution Strategies (ES, Salimans et al. [2017]). This comparison will facilitate better understanding and implementation of these algorithms. EC, ES, and NES share the same unbiased Monte-Carlo gradient estimator, gradient update approach, and fitness shaping trick, as described in Section 4.

The primary distinction between these algorithms lies in their search spaces, as illustrated in Figure S1. Both ES and NES employ real-valued parameters, akin to conventional deep learning techniques, and sample the search space using parameterized normal distributions. In contrast, EC utilizes a 1-bit discrete search space parameterized by Bernoulli distributions. This 1-bit formulation reduces the parameter size, endowing EC with unique advantages such as faster computation, improved memory efficiency, and reduced energy and cost requirements.

Additionally, there is a minor difference in how EC enhances exploration. As described in Section 4, EC clips the elements of connection probability matrix to boost exploration.

## C    Optimizing 1-bit Networks Using ES and SG

ES and SG are continuous optimization techniques designed for continuous parameters. To apply these methods to the training of 1-bit connection RSNNs proposed in this paper, we utilize discretization methods as described by Zhou et al. [2019]. Specifically, we employ ES and SG to optimize a continuous parameter $\boldsymbol{\theta}$, which is then discretized into 1-bit weights $\mathbf{W}$ by applying a threshold at 0 as $\mathbf{W} = H(\boldsymbol{\theta})$, where $H$ denotes the Heaviside step function. For SG, the straight-through estimator is utilized as a standard approach [Bengio et al., 2013]. It is crucial to acknowledge that employing continuous optimization methods in conjunction with discretization may result in biased gradient estimation, while EC naturally provides unbiased gradient estimation for discrete 1-bit connections.

## D    On Chip Power Consumption Estimation

We estimate the power consumption of EC by considering the number of spiking neuron operations and the energy per operation provided by Loihi [Davies et al., 2018], as a representative neuromorphic

Table 2: Energy consumption and network specifics

| Parameter | Value |
|---|---|
| Energy per synaptic spike op $P_s$ | 23.6 (pJ) |
| Within-tile spike energy $P_w$ | 1.7 (pJ) |
| Energy per neuron update $P_u$ | 81 (pJ) |
| # Generations $G$ | 1000 |
| # Population $P$ | 10240 |
| # Time steps $S$ | 33200 |
| # Neurons $N$ | 256 |
| # Spikes per neuron per step $R$ | 0.025 |
| # Connection per neuron $C$ | 128 |
| # Update operations per neuron $I$ | 4 |

device. The data employed for this estimation is presented in Table 2. Initially, we computed the estimated energy consumption for a single network inference as follows:

$$E_{one} = P_u * N * I * S + (P_s + C * P_w) * N * R * S = 2.8 mJ$$

Subsequently, we determined the total energy consumption during the training process:

$$E_{tot} = E_{one} * G * P = 28 kJ$$

It is important to note that these calculations are approximations and serve as an initial assessment. In future research, we plan to conduct experiments using neuromorphic chips to empirically validate the energy efficiency of RSNNs.

# E  Performance Comparison with PPO

In order to assess the efficacy of the EC-RSNN approach in relation to contemporary deep reinforcement learning (RL) techniques, we conducted additional training with GRU and LSTM models using Proximal Policy Optimization (PPO) [Schulman et al., 2017] on the most complex Humanoid task. The results of these experiments are presented in Figure S3. The data clearly indicates that EC-RSNN surpasses the performance of PPO-LSTM, illustrating a level of performance that is on par with state-of-the-art deep RL methods.

# F  Full SG Results

We present a comprehensive analysis of the Surrogate Gradient (SG) results, considering various damping factors ($\gamma$) and the parameter ($\beta$). The outcomes are illustrated in Figure S2.

# G  Detailed Experiment Settings

## G.1  Hardware

All experiments were conducted on a single GPU server with the following specifications. Additionally, all efficiency experiments, including those that measure wall-clock time, were executed on a single GPU within this server.

- 8x NVIDIA Titan RTX GPU (24GB VRAM)
- 2x Intel(R) Xeon(R) Silver 4110 CPU
- 252GB Memory

## G.2  Hyperparameters

In this section, we provide the hyperparameters for our framework, Evolving Connectivity (EC), and the baselines, including Evolution Strategies (ES) and Surrogate Gradient (SG). All methods utilize the same hyperparameter set for all three locomotion tasks. We also list the settings of neural networks below.

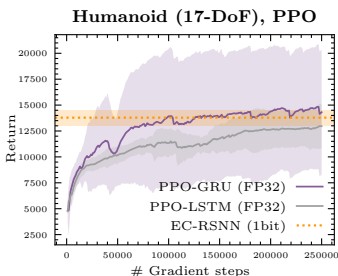

Figure S3: Performance comparison among PPO-GRU, PPO-LSTM, and EC-RSNN.

Table 3: Hyperparameters for Evolving Connectivity (EC)

| Hyperparameter | Value |
| --- | --- |
| Population size $N$ | 10240 |
| Learning rate $\eta$ | 0.15 |
| Exploration probability $\epsilon$ | $10^{-3}$ |

Table 4: Hyperparameters for Evolution Strategies

| Hyperparameter | Value | Target |
| --- | --- | --- |
| Population size $N$ | 10240 | All |
| Learning rate $\eta$ | 0.15 | RSNN |
|  | 0.01 | Deep RNN |
| Noise standard deviation $\sigma$ | 0.3 | RSNN |
|  | 0.02 | Deep RNN |
| Weight decay | 0.1 | All |

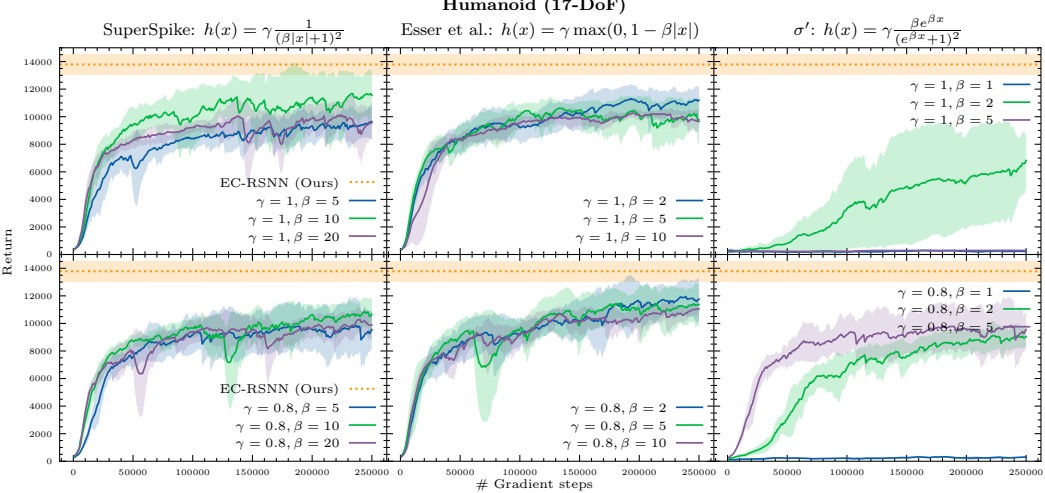

Figure S2: Surrogate Gradient on the Humanoid task, with several surrogate functions, $\beta$ and $\gamma$ parameters. Performance is sensitive to the chosen surrogate function and its parameters.

Table 5: Hyperparameters for Surrogate Gradient

| Hyperparameter | Value |
|---|---|
| *Proximal Policy Optimization (PPO)* | |
| Batch size | 2048 |
| BPTT length | 16 |
| Learning rate $\eta$ | $3 \times 10^{-4}$ |
| Clip gradient norm | 0.5 |
| Discount $\gamma$ | 0.99 |
| GAE $\lambda$ | 0.95 |
| PPO clip | 0.2 |
| Value loss coefficient | 1.0 |
| Entropy coefficient | $10^{-3}$ |
| *Surrogate Gradient* | |
| Surrogate function | SuperSpike |
| Surrogate function parameter $\beta$ | 10 |

Table 6: Settings of neural networks

| Hyperparameter | Value |
|---|---|
| *Recurrent Spiking Neural Network (RSNN)* | |
| Number of neurons $d_h$ | 256 |
| Excitatory ratio | 50% |
| Simulation time per environment step | 16.6 ms |
| Simulation timestep $\Delta t$ | 0.5 ms |
| Synaptic time constant $\tau_{syn}$ | 5.0 ms |
| Membrane time constant $\tau_m$ | 10.0 ms |
| Output time constant $\tau_{out}$ | 10.0 ms |
| Input membrane resistance [1] $R_{in}$ | $0.1 \cdot \tau_m \sqrt{\frac{2}{d_{in}}}$ |
| Hidden membrane resistance [1] $R_h$ | $1.0 \cdot \frac{\tau_m}{\tau_{syn}} \sqrt{\frac{2}{d_h}}$ |
| Output membrane resistance [1] $R_{out}$ | $5.0 \cdot \tau_{out} \sqrt{\frac{2}{d_h}}$ |
| *Gated Recurrent Unit (GRU)* | |
| Hidden size | 256 |
| *Long-short term memory (LSTM)* | |
| Hidden size | 128 |

# H  Source Code and Licensing

The source code associated with this paper is licensed under the Apache 2.0 License and is publicly available at `https://github.com/imoneoi/EvolvingConnectivity`. In our research, we incorporated locomotion tasks and the Brax physics simulator from Freeman et al. [2021], as well as the JAX framework [Bradbury et al., 2018]. Both of these resources are also released under the Apache 2.0 License.

---

[1]Resistance is set following Ding et al. [2022] to preserve variance.

