# OpenReview forum: "Evolving Connectivity for Recurrent Spiking Neural Networks"
_NeurIPS.cc/2023/Conference — NeurIPS 2023 poster_

### Official Review · Reviewer_mDuJ · 2023-06-30

**Soundness:** 3 good
**Presentation:** 4 excellent
**Contribution:** 3 good
**Rating:** 6
**Confidence:** 4

**Summary:**

This study presents an application of a previously developed approach, NES, to train RSNNs formulated based on connection probabilities. However, there are concerns regarding the ethical aspect of this work, specifically the reproducibility and proper citation of related works (see Weaknesses). Due to these reasons, it is challenging to recommend this manuscript for acceptance in its current form.

**Strengths:**

n/a


**Weaknesses:**

o	The lack of code availability at the time of review poses a significant barrier to verifying the claims made by the authors.

o	Additionally, it appears that an existing work by Stockl et al., 2021 from Maass group, which applies NES to learn RSNNs formulated in terms of probability skeleton inspired from Billeh et al., is not cited in this manuscript. This omission warrants clarification as it may influence the perceived contribution of this work.

o	Beyond the ethical issues outlined above, this paper has several limitations that need to be addressed. Firstly, the results presented are purely empirical and there is a lack of theoretical contributions. Secondly, the manuscript lacks an analysis of how varying the sample size impacts the results, which leaves unanswered questions regarding the robustness and efficiency of the proposed methodology. Lastly, it is unclear whether the authors have performed a thorough tuning of the hyperparameters, which could significantly impact the performance of the benchmarks.

**Questions:**

n/a

**Limitations:**

See weaknesses

---

> ### Author Rebuttal · Authors · 2023-08-09
>
> **[W1] Code availability**
>
> **[Response]**
>
> Thanks for your advice, but we cannot agree on code availability as the main reason to reject.
> According to the conference policy in Call for Papers, NeurIPS strongly **encourage** accompanying code and data to be submitted with **accepted papers** when appropriate, rather than a mandatory requirement.
> And, we have already promised in Appendix A that we will release the code upon the paper is accepted.
> However, to address your concern, we are glad to send a copy of the code to the Area Chair.
>
> ---
>
> **[W2] An similar existing work by Stockl et al., 2021 using NES**
>
> **[Response]**
>
> We appreciate your attention to the existing work by Stockl et al., 2021, which applies NES to learn RSNNs formulated in terms of probability skeleton inspired by Billeh et al. We acknowledge the relevance of this work and will include it in our revised manuscript.
>
> However, our EC framework has several fundamental differences from the approach taken by Stockl et al., 2021 and NES, which we detail below:
>
> 1. **Hyperparameter searching vs. parameter training**. Stockl et al. search tens to hundreds of free parameters to characterize the connection skeleton, akin to hyperparameter optimization in deep learning and HyperNEAT in evolutionary computation. In contrast, our EC framework operates more like training 1-bit neural networks in deep learning, searching every one-to-one connection parameter in the RSNN, and is agnostic to network architecture. In our experiments, we train the full 193K RSNN connections using EC, which represents more than $1000\times$ the number of dimensions than the 44 to 164 hyperparameters for the probabilistic model explored by Stockl et al.
>
> 2. **Discrete vs. continuous**. Our work presents a novel approach to the NES framework by utilizing a 1-bit discrete search space formulation, which distinguishes it from the original NES paper (xNES, SNES, Wierstra et al., 2011) and its derivatives like ES (Salimans et al., 2017). These prior works employed continuous search spaces parameterized by normal distributions to train real-valued parameters in neural networks, as seen in conventional deep learning.
>
> Our EC framework also offers several unique advantages:
>
> 1. **High performance**. Our results demonstrate that EC outperforms ES on training RSNNs, despite EC-RSNN using 1-bit discrete parameters, which are 1/32 the size of the 32-bit floating point precision employed by ES-RSNN.
>
> 2. **Faster training & inference**. As discussed in our paper, the 1-bit connections resulting from the EC framework offer potential for accelerated training and inference. In Fig. 5, we show that EC-RSNN exhibits $2 \sim 3\times$ efficiency compared to ES-RSNN.
>
> 3. **Scaling to complex tasks**. EC's ability to efficiently search the full 193K RSNN parameter space enables it to tackle complex tasks that require a larger number of network parameters. The Humanoid task is a challenging locomotion task in reinforcement learning community, typically addressed by neural networks with parameter scales ranging from 10K to 100K, as demonstrated by Salimans et al. and Freeman et al. Our EC framework can solve this task with performance comparable to deep RNNs.
>
> ---
>
> **[W3-a] Lack theoretical contributions**
>
> **[Response]**
>
> The focus of our manuscript is on the development and evaluation of the evolving connectivity (EC) framework for training RSNNs, emphasizing its hardware-friendly characteristics and performance.
> We do not consider it as a requirement to add theory to empirical experiments.
> It would be helpful if you can provide some detailed theoretical insights.
>
> ---
>
> **[W3-b] Impact of sample size on robustness and efficiency**
>
> **[Response]**
>
> In response to your concern regarding the robustness and efficiency of our proposed methodology, we have taken the following steps to address these concerns:
>
> 1. As shown in our paper, to evaluate the robustness of our method, we performed multiple trials (n=3) with different random seeds for each experiment. By reporting the average and standard deviation of the results, we demonstrate the consistency of our approach across various initial conditions.
>
> 2. To assess the generalization of our methodology, we conducted experiments using 1-bit deep RNN architectures, which further supports the adaptability of our approach to other network configurations.
>
> 3. In terms of efficiency, we have compared our method's performance with existing methods by measuring the wall-clock time, providing evidence for the improved efficiency of our approach.
>
> We acknowledge the importance of analyzing the impact of sample size on the results; However, we believe our current analysis sufficiently demonstrates the robustness and efficiency of our proposed methodology.
>
> ---
>
> **[W3-c] Hyperparameters tuning and benchmark**
>
> **[Response]**
>
> We appreciate your concern about the thoroughness of hyperparameter tuning, as it is crucial for the performance of the benchmarks. We assure that we have conducted an extensive tuning of the hyperparameters for all baseline models.
>
> 1. The PPO implementation in Brax (Freeman et al., 2021) obtains 11,300 for the Humanoid environment. Our baselines (SG-RSNN, ES-GRU, and ES-LSTM) yielded returns of 11,500, 13,000, and 15,000, respectively, outperforming the Brax benchmark.
>
> 2. As detailed in our paper, we tested various surrogate function parameters for SG-RSNN and selected the best set as our baseline.
>
> 3. We have conducted additional experiments, as in Supplement PDF Fig. S4 and S5, including PPO-LSTM, PPO-GRU, and a broader hyperparameter tuning on SG-RSNN, to confirm that our baseline is fairly constructed.
>
> Our rigorous tuning process contributed to the enhanced performance of the baselines, and we are confident in the thoroughness and effectiveness of our hyperparameter optimization.
>
> ---
>
> The detailed experimental settings and results can be found in the comment text and Supplemental PDF in the overall rebuttal.

---

> > ### Comment · Reviewer_mDuJ · 2023-08-13
> > **Response acknowledged**
> >
> > I appreciate the additional experiments and details that the authors provided to enhance the robustness, addressing many of my primary concerns. I also value the discussion contrasting with Stockl et al. While including code during the review process would have improved transparency, I trust the authors to provide a copy to the area chair, alleviating this concern. With these issues addressed, I will raise my score above the acceptance threshold.

---

> > > ### Author Response · Authors · 2023-08-14
> > >
> > > Thanks for your positive response to our rebuttal. We will try our best to make our paper more clear in the next version.

---

### Official Review · Reviewer_SW36 · 2023-07-05

**Soundness:** 2 fair
**Presentation:** 3 good
**Contribution:** 2 fair
**Rating:** 6
**Confidence:** 3

**Summary:**

Facing on the inaccurate and unfriendly limitation of current surrogate gradient-based learning methods for recurrent spiking neural networks (RSNN), this study develops the evolving connectivity (EC) framework for inference-only training. The EC framework reformulates weight-tuning as a search into parameterized connection probability distributions, and employs Natural Evolution Strategies (NES) for optimizing these distributions. The performance of the proposed EC is evaluated on a series of standard robotic locomotion tasks, where it achieves comparable performance with deep neural networks and outperforms gradient-trained RSNNs.

**Strengths:**

1.	The motivation is reasonable and the application scene is interesting.
2.	The framework of EC considering the weight reparameterization and connection evolution is reasonable.


**Weaknesses:**

1. It seems that the weight-based parameterization method and the NES method used in EC is not novel, it seems to use the method proposed in other papers.
2. The energy consumption should be computed to show the efficiency of the proposed framework.
3. The experiments seems not enough to verify the effectiveness of the proposed model.


**Questions:**

1.	The novelty of the weight-based parameterization method and the NES methods in EC framework should be explained further.
2.	The energy consumption should be computed to show the efficiency of the proposed framework.
3.	How about the performance comparison with Transformer models?


**Limitations:**

See the weakness.

---

> ### Author Rebuttal · Authors · 2023-08-09
>
> **[Q1]** *The novelty of the weight-based parameterization method and the NES methods in EC framework should be explained further.*
>
> **[Response]**
>
> Thank you for your advice.
> Our work introduces a novel approach to the NES framework by utilizing a 1-bit discrete search space formulation, which distinguishes it from the original NES paper (xNES, SNES, Wierstra et. al) and its derivatives like ES (Salimans et. al). Prior studies employed continuous search spaces parameterized by normal distributions to train real-valued parameters in neural networks, as seen in conventional deep learning.
>
> Our proposed method searches for discrete 1-bit connection-based parameterization within the NES framework, deriving a natural gradient estimator tailored for this discrete space. This approach offers several advantages over the continuous parameter search using ES. As demonstrated in our paper's experiments, our EC method yields superior final performance and accelerates the training process by 2 to 3 times. Additionally, the use of integer arithmetic for 1-bit values, as opposed to floating-point arithmetic for continuous values, reduces computational costs and enhances compatibility with neuromorphic hardware.
>
> ---
>
> **[Q2]** *The energy consumption should be computed to show the efficiency of the proposed framework.*
>
> **[Response]**
>
> Thank you for your valuable suggestion. In our study, all experiments are conducted on the same NVIDIA Titan RTX GPU, operating at a consistent 100% GPU power (280W). As a result, the energy consumption of the training process is estimated proportional to the computation wall time as shown in Table 1. Our EC framework demonstrates an energy improvement over SG and even more than 2 times to ES. We will incorporate energy consumption calculations in Section 6 to further demonstrate the efficiency.
>
> *Table 1: Estimated training power comsumption*
> | Model   | Run time (h) | Power Consumption (KWh) |
> |---------|--------------|-------------------------|
> | EC-RSNN | 18           | 5.0                     |
> | ES-RSNN | 46           | 12.9                    |
> | SG-RSNN | 20           | 5.6                     |
>
> ---
>
> **[Q3]** *How about the performance comparison with Transformer models?*
>
> **[Response]**
>
> Thank you for your question. Although Transformer models have demonstrated remarkable performance in numerous domains, their direct application to online reinforcement learning (RL) presents challenges. Specifically, conventional Transformer models encounter stability issues in RL tasks and require modifications for stabilization (Emilio Parisotto et al. 2019). We appreciate your suggestion and consider it a promising avenue for future research.

---

> > ### Comment · Reviewer_SW36 · 2023-08-11
> >
> > Thanks for the explaination and the additional experiments.
> >
> > It looks like the energy consumption of the EC-RSNN over SG-RSNN is not so advantagous, it is strange since EC is claimed to utilize a 1-bit discrete search space, please explain the reason further.
> >
> > The transformer on online reinforcement learning have been explored further in recent years, such as Zheng, Q., Zhang, A., & Grover, A. (2022, June). Online decision transformer. In international conference on machine learning (pp. 27042-27059). PMLR.

---

> > > ### Author Response · Authors · 2023-08-12
> > >
> > >
> > > **[Comment 1]** *It looks like the energy consumption of the EC-RSNN over SG-RSNN is not so advantageous, it is strange since EC is claimed to utilize a 1-bit discrete search space, please explain the reason further.*
> > >
> > > **[Response]**
> > >
> > > Thank you for your insightful comment. We appreciate the opportunity to further clarify the energy consumption comparison between EC-RSNN and SG-RSNN. There are two primary reasons for it:
> > >
> > > 1. **Comparison Standard.** Since epoch in SG and generation in EC are different concepts, our initial comparison computed the final energy consumption under the similar total computation time, which did not account for the difference in the final return. In fact, EC-RSNN achieved a higher final return (i.e., 13,808) compared to SG-RSNN (i.e., 11,505).
> > >
> > >    To provide a more comprehensive comparison of energy consumption, we have included an additional evaluation based on the energy consumption required to 'solve' the Humanoid task. We have set the 'solve' return thresholds at 11,300, which corresponds to the PPO return reported by the Brax maintainer. As illustrated in Table 1, when comparing the energy consumption at the same return thresholds, EC-RSNN demonstrates a more advantageous energy efficiency than SG-RSNN. This comparison highlights the benefits of utilizing a 1-bit discrete search space in EC-RSNN and provides a clearer understanding of the energy consumption differences between the two models.
> > >
> > > |   Model   | Return | Run time (h) | Power Consumption (KWh) |
> > > |:---------:|:------:|:------------:|:-----------------------:|
> > > |  EC-RSNN  | 11,300 |      6       |           1.7           |
> > > |  SG-RSNN  | 11,300 |     20       |           5.6           |
> > >
> > > 2. **GPU Implementation.** It is noteworthy that GPUs have significantly expedited floating-point operations, a prevalent data type within the deep learning community. However, 1-bit is not a built-in data type in JAX and GPU, like most computational platforms.
> > >
> > >    In our current JAX code, we could only utilize the boolean data type as a substitution, whose efficiency is limited compared to a 1-bit implementation. Despite this implementation difficulty, we achieved 3x efficiency with the same return. It demonstrated that our EC-RSNN can have a potentially more significant advantage compared to SG-RSNN. In the future, we will further try to implement our EC-RSNN on specially designed neuromorphic devices supporting 1-bit implementation.
> > >
> > > ---
> > >
> > > **[Comment 2]** *The transformer on online reinforcement learning has been explored further in recent years, such as Zheng, Q., Zhang, A., & Grover, A. (2022, June). Online decision transformer. In the international conference on machine learning (pp. 27042-27059). PMLR.*
> > >
> > > **[Response]**
> > >
> > > Thanks for your insightful suggestion. The paper you referenced blends offline pretraining and online finetuning; it does not entirely adopt the framework of online learning. Actually, offline reinforcement learning has become a prevailing approach within the domain of transformer models (Lili Chen et al., 2021; Michael Janner et al., 2021). In contrast, online learning remains an area of ongoing research, primarily due to the persisting challenges related to stability (Emilio Parisotto et al., 2019).
> > >
> > > It is worth highlighting that all of our conducted experiments are grounded in the paradigm of online learning, thus rendering direct comparisons with pre-trained transformers incongruous. Despite this distinction, we intend to explore it and run additional experiments in the future.

---

> > ### Comment · Reviewer_SW36 · 2023-08-13
> >
> > Thanks for your responses.
> >
> > I appreciate the addidtional experiments, however, the power consumption is not compared with ANN models, it makes it not convinced to show the energy efficiency advantage of SNNs. Is there any other ways to compare the energy consumption for the comparison?

---

> > > ### Author Response · Authors · 2023-08-15
> > >
> > > Thank you for your question.
> > > To further demonstrate our energy efficiency, we have conducted computations pertaining to the estimated energy consumption of EC-RSNN when implemented on the Loihi chip [1]. The data we used are presented in the table below.
> > >
> > > *Table: Energy data from Loihi [1] & Network data from EC-RSNN*
> > > | Parameter                             | Value     |
> > > |---------------------------------------|-----------|
> > > | Energy per synaptic spike op $P_s$    | 23.6 (pJ) |
> > > | Within-tile spike energy $P_w$        | 1.7 (pJ)  |
> > > | Energy per neuron update $P_u$        | 81 (pJ)   |
> > > | # Generations $G$                     | 1000      |
> > > | # Population $P$                      | 10240     |
> > > | # Time steps $S$                      | 33200     |
> > > | # Neurons $N$                         | 256       |
> > > | # Spikes per neuron per step $R$      | 0.025     |
> > > | # Connection per neuron $C$           | 128       |
> > > | # Update operations per neuron $I$    | 4         |
> > >
> > > Firstly, we calculated the estimated energy consumption of one network.
> > >
> > > $$
> > > E_{one} = P_u * N * I * S + (P_s + C * P_w) * N * R * S = 2.8mJ
> > > $$
> > >
> > > Then, we calculated the total energy consumption during training.
> > >
> > > $$
> > > E_{tot} = E_{one} * G * P = 28 kJ
> > > $$
> > >
> > > Our ANN baselines (ES/PPO-LSTM/GRU) demand several hours of training time when executed on GPUs, consequently leading to energy consumption on the order of megajoules (MJ).
> > > Our calculations, however, indicate a noticeable reduction in power consumption by an order of magnitude of approximately $1\sim2\times$.
> > > Please note that these computations are estimates and serve as a preliminary evaluation.
> > > In the future, we intend to conduct experiments using neuromorphic chips to empirically substantiate the disparity in energy consumption between EC-RSNN and RNNs.
> > >
> > > [1] Loihi: A Neuromorphic Manycore Processor with On-Chip Learning, 2018.

---

### Official Review · Reviewer_ho2W · 2023-07-05

**Soundness:** 4 excellent
**Presentation:** 4 excellent
**Contribution:** 4 excellent
**Rating:** 8
**Confidence:** 5

**Summary:**

The paper describes an evolutionary training algorithm to optimize the binary weight matrix of a recurrent spiking neural network. The optimization algorithm is derived using natural evolutionary strategies assuming that the weights follow a Bernoulli distribution with parameter $\rho$. The resulting weight update is remarkably simple, and despite that, it seems to work very well.

The algorithm is tested on the classical locomotion tasks of simulated robotics, and the performance is reported in terms of task return and time to solution in wall clock time. On all the task RSNN model trained in this way performs better or similar to the LSTM or GRU baselines trained with ES, and it is always better than the spiking networks trained with surrogate gradient. By leveraging the discrete values of the weight matrix, the implementation is also faster than a floating point evolutionary strategy.

**Strengths:**

As said in the paper: inference-only training solutions can be very useful for hardware where the training algorithm cannot be implemented easily. This approach is likely to have important for training or fine-tuning architectures when deployed on unconventional devices.

 I would also highlight an important achievement that the authors did not mention enough in my opinion: it appears that this training algorithm is better than all existing training algorithms for spiking networks which is remarkable for something that is practicable and competitive with deep learning solutions to such problems. It could in fact provide opportunities for quantized and spiking architectures which are otherwise bounded in their performance by the approximate gradients that are usually computed with surrogate gradient and straight through. This is solved here since the gradients are unbiased (not mentioned by the authors as far as I see). This was probably why surrogate gradient baselines do not quite reach the performance of LSTM and GRU baselines as seen here as in previous papers, but the algorithm provides a solution to this problem.

**Weaknesses:**

1) There are a few data points that would be useful to fully evaluate the quality of the training algorithm and separate the benefits of the spiking architecture from the benefits of the algorithm.
a. Is it possible to train an LSTM or a GRU with EC? What is its return after training, is that also better than ES for the spiking neurons?
b. What is the return obtained with LSTM / GRU and PPO? This is both important to verify that PPO is well-implemented and claim or not that the EC-RSNN is competitive with the regular deep learning approach. (If the performance is low, then maybe there is a problem with the surrogate implementation too).

2) I think it would be valuable to clarify that most surrogate gradient techniques were typically tested mainly without recurrent connections (Slayer, super-spike for instance). I find it admirable that the authors tested so many baselines already, but also unfortunate that they might have missed important details that were reported the be crucial in the presence of recurrent connections: depending on the weight initialization it was shown that the pseudo derivate should be multiplied by a dampening factor < 1 to avoid the accumulation of approximation errors, with Glorot initialization you might be fine without this (see Celotti and Rouat 2022). But what initialization did you use and did you test this?

3) I realized only later that most of the concrete hyperparameters are detailed in the appendix: How many layers, units per layer, batch size, epsilon. Please state clearly in the main text that all this important is available in the appendix when it's relevant. I find it in general even better to put directly these numbers in the main text when possible. (For instance, write $\epsilon = 0.001$ instead of $\epsilon \rightarrow 0$). Same remark when talking about the hardware. Instead of writing "GPGPU" or "identitical hardware" which I find vague since neuromorphic hardware is often mentioned in the main text. When possible, I would encourage writing just NVIDIA GPU and pointing out the appendix where the hardware reference is provided. On a related note, I would also encourage the authors to prefer a specific reference to the Figure number and panel at line 277 when commenting on their results.

4) In the context of this conference which is broadly addressed to the machine learning community, I think it is important that the surrogate gradient is nothing more than a variant of the straight-through gradient estimator from Bengio 2013 which is much better known outside of the neuromorphic community and was discovered before.

5) I find the first sentence line 20 a bit strange, and I find it weird to reference Pei et al. at this point. AGI is also never used elsewhere in the text (for good reasons since you have more concrete and good results to discuss), so why defining this acronym here ?

**Questions:**

I realized that points 1a.b and 2 above would be best addressed with extra simulations. Given the time and effort that it requires, I do not consider those simulations necessary, but I would encourage the authors to address those questions verbally during the rebuttal and consider the feasibility and the cost/benefits of these potential simulations at some point.

For points 3, 4 and 5. I would encourage the authors to do minor edits.

More general question: Do the authors believe that this approach can replace gradient descent learning in some other context ? Or is it that ES and EC and bounded to make only sense in this type of toy reinforcement leanring task where the network size is small and the batch size can be super large?



**Limitations:**

I see no potential negative impact. The limitations are clearly stated in the paper.

---

> ### Author Rebuttal · Authors · 2023-08-09
>
> **[S1]** *The gradients of the EC training algorithm are unbiased.*
>
> **[Response]**
>
> Thank you for your affirmation and suggestion.
> We have also pointed out that "the surrogate gradient leads to inherent inaccuracy in the descent direction" in line 37.
> We will further emphasize the contribution of our unbiased gradient property in the introduction of our revised manuscript.
>
> ---
>
> **[W1-a]** *Is it possible to train an LSTM or a GRU with EC? What is its return after training? Is that also better than ES for spiking neurons?*
>
> **[Response]**
>
> Thank you for your insightful question. While training an LSTM or GRU with EC is theoretically possible, it requires scaling and tweaking the gating mechanism to adapt to 1-bit weights (Xuan Liu et al., 2018). For simplicity, we conduct experiments using a vanilla RNN trained with EC.
>
> As a result, we obtained 12,455 returns with EC-RNN, compared to 11,042 returns with ES-RNN and 11,133 returns with PPO-RNN. Detailed results can be found in Fig. S3 in the supplementary PDF.
> The results demonstrate that EC can also train 1-bit deep recurrent neural networks, showcasing its potential for different architectures and quantized neural networks. We will add these experiment results in Section 6.
>
> ---
>
> **[W1-b]** *What is the return obtained with LSTM / GRU and PPO?*
>
> **[Response]** Thank you for your question.
>
> We train PPO-LSTM and PPO-GRU with the 12,960 and 14,312 average final return, while Brax (Freeman et al., 2021) report a PPO return of approximately 11,300.
> As a result, our EC-RSNN achieves a comparable final return (i.e., 13,808) with PPO, demonstrating competitive performance with deep reinforcement learning. More detailed results can be found in Fig. S4 of the Supplemental PDF.
> We will add these experiment results in Section 6.
>
> ---
>
> **[W2]** *Miss important details that were reported the be crucial in the presence of recurrent connections: Depending on the weight initialization it was shown that the pseudo derivate should be multiplied by a dampening factor $< 1$ to avoid the accumulation of approximation errors. But what initialization did you use and did you test this?*
>
> **[Response]** Thanks for your advice. We conduct further hyperparameter tuning for our SG-RSNN baseline, introducing a dampening factor equal to $0.8$. Our experiment shows a similar maximum performance (i.e., 11,765) with the SG-RSNN baseline (i.e., 11,546) from our paper. The detailed results can be seen in Fig. S5 in the supplementary PDF. We will replace this experiment setup in Section 6.
>
> In terms of initialization, all parameters are initialized to $0.5$, as a balance point between connection and no connection in the Bernoulli distribution. For a fair initialization, we moved the $1/\sqrt N$ term in LeCun Normal initialization to a product of hyperparameters, as shown in Table 1 below. We will add the initialization in Section 4 in our revised paper.
>
> *Table 1: Initialization*
>
> | Hyperparameter                     | Value                                          |
> |------------------------------------|------------------------------------------------|
> | Input membrane resistance $R_{in}$   | $0.1 * \tau_m * \sqrt{2/d_{in}}$                   |
> | Hidden membrane resistance $R_h$   | $1.0 * \tau_m / \tau_{syn} * \sqrt{2/d_{h}}$          |
> | Output membrane resistance $R_{out}$ | $5.0 * \tau_{out} * \sqrt{2/d_{h}}$                  |
>
> ---
>
> **[Q2]** *For weaknesses points 3, 4 and 5. I would encourage the authors to do minor edits.*
>
> **[Response]** Thank you for your valuable suggestions. In response to [W3] and [W4], we will move the main information about hyperparameters and hardware to the main text.
> Meanwhile, we will further polish our statements to be more clear.
>
> As for [W5], we recognize that the reference to AGI in line 20 may seem out of place. In our revised manuscript, we will provide a more appropriate introduction to neuromorphic computing, ensuring a clear and concise presentation of the topic.
>
> ---
>
> **[Q3]** *Do the authors believe that this approach can replace gradient descent learning in some other context? Or is it that ES and EC and bounded to make only sense in this type of toy reinforcement learning task where the network size is small and the batch size can be super large?*
>
> **[Response]**
>
> Thanks for posing this thought-provoking question.
>
> Firstly, humanoid locomotion is a challenging task in reinforcement learning (RL), as it has high dimensional observation and action space (Duan et al., 2016), and poses a challenge to deep RL algorithms (Haarnoja et al., 2018).
> Extending beyond locomotion tasks, we believe that the EC framework has the potential to be useful in other contexts, such as image classification (Xingwen Zhang et al., 2017) and game playing (Salimans et al., 2017), where evolutionary algorithms have demonstrated success.
>
> Secondly, under large batch size conditions, our EC is much more efficient than ES and SGD.
> On the one hand, our EC can be efficiently distributed over multiple devices by sending random seeds, while SGD requires computing an average of the gradient of all batches incurring significant communication costs.
> On the other hand, the 1-bit connections in the EC framework may reduce memory and computation costs to 1/32 compared to the FP32 commonly used by both ES and SGD.
> Moreover, EC requires only a forward pass, while SGD needs both a forward and backward pass.
> Namely, the 10,240 batch size in EC is analogous to $\frac{10240}{2 \times 32} = 160$ in SGD in memory usage, which is comparable to commonly-used SGD batch sizes such as 128 and 256.
>
> In summary, we believe that the EC framework holds promise for a broader range of applications and is capable of training larger networks with larger batches.
> We will explore different tasks and the efficiency of the EC framework in future work.

---

> > ### Comment · Reviewer_ho2W · 2023-08-11
> > **Thank you**
> >
> > Thank you for clarifying things and running additional simulations. I find indeed that the comparison between PPO-RNN and EC-RNN are compelling. These are great additions for the paper, the choice of dampening factor 0.8 is not necessarily standard (I had seen 0.3 for initialization $1/\sqrt{n}$) but the results in the non spiking RNN suggest that the stability of the gradient estimator would not explain the failure of PPO, so I am now convinced.
> >
> > I had forgotten one question, do you subtract the mean or any baseline to the return R in equation (14). I believe this should yield a gradient estimator with lower variance? If not, do you know why is it not necessary?

---

> > > ### Author Response · Authors · 2023-08-12
> > >
> > > Thanks for your professional question.
> > >
> > > Alternatively, we adopted a similar fitness shaping trick, center rank transform, as discussed in ES (Tim Salimans et al., 2017) and NES (Daan Wierstra et al., 2014). The transform sort the return of a population, then rescale the sorting indexes into a fixed interval $[-0.5, 0.5]$. Evidently, it yields an outcome with zero mean and fixed variance. Moreover, it reduces the influence of the outlier individual and helps with stability. We will include the details in our revised paper.
> > >
> > > Additionally, during the research process, we tried several other fitness shaping methods on population return, including no rescaling, subtraction of mean, and rescaling to fixed interval $[-0.5, 0.5]$. We found out that these methods can train EC, but resulting in a lower final return and often come with stability issues.

---

> > > > ### Comment · Reviewer_ho2W · 2023-08-13
> > > >
> > > > Thank you for the clarification, yes I think adding these details would be useful.
> > > >
> > > > Altogether, I see no flaws in the paper and the new simulations and clarifications are improving the paper quality. I therefore increase my rating by one point.

---

> > > > > ### Author Response · Authors · 2023-08-14
> > > > >
> > > > > Thanks for your affirmation and constructive suggestions. We will carefully improve our paper according to your advice.

---

### Official Review · Reviewer_Apb4 · 2023-07-09

**Soundness:** 3 good
**Presentation:** 3 good
**Contribution:** 3 good
**Rating:** 7
**Confidence:** 3

**Summary:**

The authors present a new evolutionary algorithm, evolving connectivity (EC) to train recurrent spiking neural networks (RSNN). The algorithm alleviates the gradient estimation problem of surrogate gradient-based (SG) training methods which are often hard to implement in neuromorphic hardware. Further, by focusing the algorithm only on evolving network connectivity instead of the weight magnitude, EC can reduce all network weights down to 1bit {0,1} alternatives making it further hardware friendly. The paper also highlights the superior accuracy and performance of their training algorithm for a few robotics tasks involving sequential decision making against state-of-the-art Surrogate Gradient and Evolution Strategies based algorithms.

**Strengths:**

The authors present a strong argument for developing RSNNs which focus more on the connectivity probability between layers instead of selecting specific weight magnitudes. This enables the network to implicitly fit many samples without overfitting to any specific subset of data.

The paper explains the EC algorithm clearly and highlights justifications for its superiority in performance and accuracy compared to other algorithms (ES/SG) for targeted robotics benchmarks.

Removing the gradient estimation requirement and reducing the weight matrix to 1-bit are significantly useful optimizations that make the algorithm hardware friendly.


**Weaknesses:**

The paper doesn’t clearly describe the RSNN architecture that is adopted for the experiments discussed and how it compares in terms of number of neurons/connections to the Deep RNNs that are used as baseline for performance/accuracy. This makes the comparisons to the ES-RNN results difficult to understand for me.

Further, comparing the RSNN based experiments only (ES/EC/SG), it appears that only EC algorithm is 1-bit owing to the focus on connectivity only. But this makes me wonder if 1-bit networks applying ES/SG algorithms could compete against EC approach in terms of performance. I understand that by the authors’ assertion that connectivity is more critical than assigning wide ranging weights to a fixed set of connections. However, for experiment’s sake I am curious to understand that if ES/SG algorithms also had the HW friendly assumption of only creating a binary SNN then would their accuracy/performance improve or deteriorate?


**Questions:**

1.	Describe the RSNN architecture and how it compares to the RNNs used in Experiments in terms of number of layers/params/model size/precision?
2.	Experiment with 1-bit RSNN utilizing ES/SG algorithms for training to understand the impact on accuracy and performance?
3.	The combination of ES and RSNN performs notably well on all the benchmarks tested, but it would be good to understand what part of it comes from the network architecture (try another architecture), what part comes from bit precision and lastly what is contributed by the training algorithm? This kind of study will help strengthen the credibility of EC algorithm for other benchmarks and network architectures.

**Limitations:**


The authors provide a useful discussion on the memory footprint of the different training algorithms discussed. Please address the concerns related to network architecture and precision impacting accuracy and performance as discussed in weakness section.

---

> ### Author Rebuttal · Authors · 2023-08-09
>
> **[Q1]** *Describe the RSNN architecture and how it compares to the RNNs used in Experiments in terms of number of layers/params/model size/precision?*
>
> **[Response]**
>
> Thanks for your valuable question.
> The table below outlines the similarities and differences between RSNN and baseline models in terms of the number of layers, hidden size, number of parameters, precision, and model size.
> To ensure a fair comparison, our RSNN (EC) architecture has a similar input-hidden-output structure and number of parameters to the RNNs used in the experiments.
> The primary difference lies in the precision, with RSNN (EC) utilizing 1-bit precision, while other baseline models employ FP32 precision.
> Consequently, RSNN (EC) has a smaller model size.
> In our final version, we will include this table in Section 6.1 and further elaborate on the architecture information.
>
> *Table 1: Comparison of Model Architectures*
> | Model         | Hidden size | # Hidden Layers | # Params | Precision | Size (KB) |
> |---------------|-------------|-----------------|----------|-----------|-----------|
> | RSNN (EC)     | 256         | 1               | 193K     | 1-bit     | 24        |
> | RSNN (Others) | 256         | 1               | 193K     | FP32      | 768       |
> | GRU           | 256         | 1               | 386K     | FP32      | 1544      |
> | LSTM          | 128         | 1               | 191K     | FP32      | 764       |
>
> ----
> **[Q2]** *Experiment with 1-bit RSNN utilizing ES/SG algorithms for training to understand the impact on accuracy and performance?*
>
> **[Response]**
>
> Thanks for your advice. Despite ES and SG being continuous optimization methods built for continuous parameters, we attempted to discretize ES and SG for training 1-bit connection RSNNs proposed in our paper.
> Specifically, we adopt ES and SG to optimize a continuous parameter $\theta$ and discretize them to 1-bit weights $\textbf{W}$.
> We use threshold at 0 as $\textbf{W}=H({\theta})$, where $H$ is the Heaviside step function. For SG, we additionally used the straight-through estimator as a common practice (Bengio et al. 2013).
> Please note that these approximations and discretization may result in biased gradients for the 1-bit discrete optimization.
>
> The results in Table 2 show that ES-RSNN (1-bit) and SG-RSNN (1-bit) exhibit learning progress but still remain a significant gap to EC-RSNN (1-bit). Besides, ES and SG performed better on continuous FP32 weights compared to discrete 1-bit. This suggests that continuous optimization methods excel with continuous parameters.
> For 1-bit discrete connections, EC should be adopted as it is specifically designed for discrete 1-bit optimization and provides unbiased gradients. Detailed results can be found in Fig. S3 in the supplementary PDF.
> We will add these experiment results in Section 6.
>
> *Table 2: Comparison of Model Architectures*
> | Algorithm | Precision | Return |
> |-----------|-----------|--------|
> | EC-RSNN   | 1-bit     | 13808  |
> | ES-RSNN   | 1-bit     | 10240  |
> | SG-RSNN   | 1-bit     | 6067   |
> | ES-RSNN   | FP32      | 11264  |
> | SG-RSNN   | FP32      | 11505  |
>
> ----
> **[Q3]** *Network architecture and precision impact accuracy and performance.*
>
> **[Response]**
>
> Thanks for your suggestion. To validate the EC on more network architectures, we conducted experiments using a vanilla RNN trained with EC, meanwhile using ES and PPO as a baseline. The results can be seen in Table 3 and Figure S2 in the supplementary PDF. It demonstrates that EC can effectively train 1-bit deep recurrent neural networks and has the potential for different architectures and quantized neural networks.
>
> *Table 3: Vanilla RNN*
> | Algorithm | Precision | Return |
> |-----------|-----------|--------|
> | EC-RNN    | 1-bit     | 12455  |
> | ES-RNN    | FP32      | 11042  |
> | PPO-RNN   | FP32      | 11133  |
>
> Regarding the impact of precision, Table 2 highlights that ES and SG, not specifically designed for 1-bit precision, encounter optimization challenges, which negatively affect performance. In contrast, EC, designed to optimize 1-bit connections, is not hindered by these challenges and outperforms the RSNN baselines. This evidence supports the robustness and adaptability of the EC approach across varying network architectures and precision levels.

---

> > ### Comment · Reviewer_Apb4 · 2023-08-16
> > **Response to Author Rebuttal**
> >
> > Thanks to the authors for providing detailed feedback on the questions discussed in the review.
> >
> > A few comments on the response:
> >
> > 1. The breakdown of network parameters and memory footprint is very helpful.
> >
> > 2. I appreciate the effort to try out the suggested experiment, I do see a significant degradation in the performance of SG baseline on moving to the 1-bit precision. While the degradation for ES baseline is smaller, its performance is still lagging behind the proposed EC. Also this seems to highlight the difference in the optimization approach (continuous variable vs connectivity) and their performance for the same task.
> >
> > 3. Very useful data on Vanilla RNN training as well, thanks for generating it.
> >
> > In conclusion, I am satisfied with the authors responses and will increase my score by one point.

---

> > > ### Author Response · Authors · 2023-08-20
> > >
> > > Thank you for your thoughtful feedback. We will take your valuable suggestions into consideration and refine our paper accordingly.

---

### Author Rebuttal · Authors · 2023-08-09

Thanks for all your suggestions. In light of them, we gathered commonly asked questions and ran a comprehensive set of additional experiments, as detailed below. The result figures are shown in the supplementary PDF. All our additional experiments are configured using the same settings and hyperparameters as in our paper and Appendix A, tested on the most complex Humanoid task (17-DoF) in Brax (Freeman et al., 2021), and averaged over 3 independent random seeds, with standard deviation shown as shaded areas.

1. **EC vs. NES?** The primary distinction is discrete vs. continuous, as demonstrated in Fig. S1. Our work presents a novel approach to the NES framework by utilizing a 1-bit discrete search space formulation, which distinguishes it from the original NES paper (xNES, SNES, Wierstra et al., 2011) and its derivatives like ES (Salimans et al., 2017). These prior works employed continuous search spaces parameterized by normal distributions to train real-valued parameters in neural networks, as seen in conventional deep learning.

2. **Can EC work on other 1-bit networks?** To verify EC's capabilities for training deep RNNs, we conducted experiments using a vanilla RNN trained with EC, meanwhile trained with ES and PPO as baselines. The RNN has 256 tanh units in the hidden layer. For 1-bit EC, the weight magnitudes are 0-1 connection matrix multiplied by LeCun initialization standard deviation, and the weight signs are determined by $+$ for excitatory, $-$ for inhibitory, with the first 128 neurons as excitatory and last 128 as inhibitory. For ES and PPO as baselines, we use real-valued weights with LeCun normal initialization. The results in Fig. S2 demonstrate that EC can effectively train 1-bit deep recurrent neural networks and has the potential for different architectures and quantized neural networks.

3. **Can ES or SGD/SG work on 1-bit RSNN?** Despite that ES and SG are continuous optimization methods built for continuous parameters, we attempted to discretize ES and SG for training 1-bit connection RSNNs proposed in our paper. Specifically, we adopted ES and SG to optimize a continuous parameter $\theta$ and discretize them to 1-bit weights $\textbf{W}$ by thresholding at 0 as $\textbf{W}=H(\theta)$, where $H$ is the Heaviside step function. For SG, we used the straight-through estimator as a common practice (Bengio et al., 2013). Please note that these approximations and discretization may result in biased gradients for the 1-bit discrete optimization.

    Fig. S3 (a) shows that ES-RSNN (1bit) and SG-RSNN (1bit) exhibited learning progress but were outperformed by EC-RSNN (1bit). Figs. S3 (b) and (c) indicate that ES and SG performed better on continuous FP32 weights compared to discrete 1-bit. This suggests that continuous optimization methods excel with continuous parameters, and for 1-bit discrete connections, EC should be adopted as it is specifically designed for discrete 1-bit optimization and provides unbiased gradients.

4. **Is PPO correctly implemented?** To ensure our PPO implementation's accuracy, we trained PPO-LSTM and PPO-GRU and compared with EC-RSNN (Figure S4). The Brax maintainers reported a PPO return of approximately 11,300. Our PPO implementation yielded higher results, with 12,960 (PPO-LSTM) and 14,312 (PPO-GRU) average final return, confirming its correctness and performance. Notably, EC-RSNN achieved a higher final return than PPO-LSTM, demonstrating competitive performance with deep reinforcement learning.

5. **Is SG correctly implemented?** We conducted additional experiments using different dampening factors to thoroughly test our surrogate gradient baseline, as demonstrated in Figure S5. With a dampening factor of $\gamma=0.8$, our experiment showed a similar maximum performance to the SG-RSNN baseline from our paper.

In conclusion, our additional experiments reinforce our findings that EC is an effective novel training framework for RSNNs and that our baselines are robustly constructed.

---

### Decision · Program_Chairs · 2023-09-21

**Decision:**

Accept (poster)

**Comment:**

The authors propose an algorithm to optimize recurrent spiking neural networks (RSNNs) called evolving connectivity (EC), employing natural evolution strategies (NES) to optimize connectivity probabilities. Using this technique, weight values can be reduced to be binary. This avoids the use of surrogate gradient methods. It is evaluated on standard robotic locomotion tasks.

There were some concerns regarding the novelty of the work in the initial reviews. However, the authors could convince the reviewers that their approach provides a significant novel contribution compared to previous work. In general, the reviewers were convinced that the work presents a very interesting algorithm that is quite different form gradient-based approaches and shows excellent performance on the considered tasks. The potential benefit for neuromrophic implementations was also acknowledged.